# Effects of Different Voided Urine Sample Storage Time, Temperature, and Preservatives on Analysis with Multiplex Bead-Based Oncuria Bladder Cancer Immunoassay

**DOI:** 10.3390/diagnostics15020138

**Published:** 2025-01-09

**Authors:** Sunao Tanaka, Kaoru Murakami, Toru Sakatani, Riko Lee, Wayne Hogrefe, Fernando Siguencia, Charles J. Rosser, Hideki Furuya

**Affiliations:** 1Samuel Oschin Comprehensive Cancer Center, Cedars-Sinai Medical Center, Los Angeles, CA 90048, USA; sunao.tanaka@cshs.org (S.T.); kaorum@kuhp.kyoto-u.ac.jp (K.M.); sktntoru5@kuhp.kyoto-u.ac.jp (T.S.); riholee@hawaii.edu (R.L.); wfsiguencia@gmail.com (F.S.); or rosser@nonagen.com (C.J.R.); 2Nonagen Bioscience Corp., Los Angeles, CA 90010, USA; hogrefeconsulting@gmail.com

**Keywords:** Oncuria, bladder cancer, urine, storage

## Abstract

**Background/Objectives:** Urinalysis accuracy requires reliable sample stability that is dependent on the chosen collection and storage conditions. The multiplex Oncuria bladder cancer immunoassay currently needs urine samples stored at 4 °C until analysis, which requires more effort, equipment, and workflow than storing samples at room temperature. Thus, successful sample storage at room temperature (20 °C) may reduce laboratory handling time and expenses. This study evaluated whether different voided urine sample collection and storage parameters affected subsequent biomarker analysis with Oncuria. The Oncuria simultaneously quantifies 10 protein analytes in urine to generate a bladder cancer diagnostic signature. **Methods**: Samples were stored at varied temperatures (20 °C, 4 °C, −20 °C) for up to 1 month. The effects of adding two commercial urine sample stabilizers and antibiotics (trimethoprim) were also assessed. Subsequently, multiple potential biospecimen stabilizers were tested in urine samples and evaluated with Oncuria in hopes of allowing the urine sample to remain at room temperature for extended periods of time. **Results**: First, it was demonstrated that voided urine samples stored at room temperate without such stabilizers had different levels of the 10 analytes associated with the Oncuria test compared to voided urine samples stored at 4 °C. Next, we evaluated the effects of commercially available biospecimen stabilizers. Despite the addition of these stabilizers, the levels of the 10 analytes were altered when the samples were stored at room temperature for prolonged periods of time. Therefore, we could not identify a suitable biospecimen stabilizer that would not require sample refrigeration. **Conclusions**: To minimize sample degradation/alteration after collection, voided urine samples should be refrigerated until analyzed with Oncuria as the refrigeration is advantageous for the storage and the transport of these urine samples.

## 1. Introduction

Analysis of voided urine samples provides a non-invasive, cost-effective, easily replicated means to evaluate biomarker signatures associated with bladder cancer [1]. These biomarker profiles (e.g., protein, DNA, RNA) have value in initial bladder cancer diagnosis, tracking disease progression, and monitoring effects of medical interventions. However, urine presents a harsh environment with varying pH, high salt levels, and potential microbial contaminants, which over time could affect the levels of analytes within urine samples [2,3]. Thus, assay accuracy requires optimized urine sample storage conditions that retain the specimen’s starting biochemical profile.

Oncuria^®^ (Nonagen Bioscience Corp., Los Angeles, CA, USA) is a multiplex immunoassay that coordinately evaluates the concentrations of 10 bladder cancer biomarkers in voided urine samples, for the detection and management of bladder cancer. The concentrations of the 10 protein analytes are incorporated into a weighted logistic regression algorithm that outputs a risk score from 0 to 1.0 (with 1.0 being the highest risk). The test has undergone extensive analytical and clinical validation [4,5,6]. Currently, Oncuria requires freshly voided urine samples to be stored at 4 °C (i.e., under refrigeration) until analysis. A previous study that evaluated the stability of urine specimens for traditional urinalyses showed that keeping specimens at room temperature (i.e., ≈20 °C) requires sample pretreatment with suitable preservatives [7]. Based on previous studies [8,9], we expect that keeping the sample in an uncontrolled temperature without preservatives results in a reduction in data quality. However, the practical problem is that refrigerating urine samples requires additional labor, equipment, and workflow considerations than storing samples at room temperature and may be challenging in some settings, particularly when samples must be transported or if the laboratory is located in an austere environment. In addition, the cost for the refrigerated shipment of samples is more expensive than those for ambient shipment because of the weight due to the refrigerated container and ice packs. Thus, we are interested in assessing the impact of various urine collection and storage conditions on sample quality retention. This experimental series tested the influence of the time and temperature of urine sample storage, and the effect of adding different preservatives, on the accuracy of the Oncuria multiplex bladder cancer assay.

## 2. Materials and Methods

*Ethical Oversight:* This study received IRB approval from the Cedars-Sinai Medical Center Institutional Review Board, Los Angeles, CA, USA (IRB #00002423), with a waiver of consent. This study’s performance complied with the tenets of the Declaration of Helsinki.

*Study 1—Effect of Storage Temperature and Duration*: Urine samples were collected from five healthy subjects, centrifuged at 1000× *g* for 5 min, and pooled at the same ratio for each subject. Because urine presents a harsh environment, we expected that the pooled urine could cover potential variabilities of individual urines. The pooled urine was stored at room temperature (20 °C) for 0.5, 1, 2, 4, 8, and 24 h; at 4 °C for 24 and 48 h and 1 week; or at −20 °C for 1 day, 1 week, and 1 month. After the indicated times, samples were stored at −80 °C until analyzed with the Oncuria multiplex assay. A pooled urine sample that had been immediately stored at −80 °C was used as a control.

*Study 2—Effect of Commercial Urine Stabilizers:* Urine samples were collected from two healthy subjects and treated with one of two commercially available urine stabilizers, either the cOmplete™ Protease Inhibitor Cocktail (Sigma-Aldrich, St Louis, MO, USA) or Alere NMP22 Urine Collection Kit (Alere Inc., Waltham, MA, USA). Control urine samples had no stabilizers added. Urine samples were stored at either room temperature or 4 °C for 1, 4, 8, and 24 h. After the above times, the samples were stored at −80 °C until analyzed. Control urine was stored at −80 °C immediately after collection.

*Study 3—Effect of a Commercial Urine Preservation Tube Collection:* Urine samples were collected from two healthy subjects and aliquoted into both a standard sterile 15 mL conical polystyrene tube (control) and a Urine Collection and Preservation Tube (Cat# 18120, Norgen Biotek, Thorold, ON, Canada). Tubes were stored at either room temperature or at 4 °C for 1, 4, 8, and 24 h before storage at −80 °C until analyzed. Control urine was stored at −80 °C immediately after collection.

*Study 4—Effect of Antibiotic Addition:* Urine samples were collected and pooled from two healthy subjects. Pooled urine was aliquoted into 3 tubes: a general 15 mL conical tube (control), a conical tube containing trimethoprim (antibiotic to suppress bacterial overgrowth; final concentration of 20 μL/mL), and a Urine Collection and Preservation Tube (Norgen). Urine samples were stored at either room temperature or at 4 °C for 1, 4, 8, and 24 h. After the indicated times, samples were stored at −80 °C until analyzed.

*Multiplex Testing:* Oncuria is a custom Luminex^®^ Performance Assay (i.e., multiplex bead-based immunoassay in 96-well format) designed to detect 10 protein biomarkers to generate a bladder cancer diagnostic signature. The 10 biomarkers are the following: alpha-1 antitrypsin, A1AT; angiogenin, ANG; apolipoprotein E, APOE; carbonic anhydrase 9, CA9; interleukin 8, IL8; matrix metallopeptidase 9, MMP9; matrix metallopeptidase 10, MMP10; plasminogen activator inhibitor 1, PAI-1; syndecan 1, SDC1; and vascular endothelial growth factor, VEGF. The assay was performed as previously described [5,6]. Briefly, urine samples were thawed on ice and reacted with a microbead cocktail consisting of 10 protein analyte-specific capture antibodies, each pre-coated onto a magnetic bead set containing a unique fluorescent internal label (Luminex Corp., Austin, TX, USA). The bead mixture containing the captured analytes was then immobilized with a magnet, washed, and incubated with a detection antibody cocktail comprising a set of 10 analyte-specific biotinylated antibodies. Following another round of magnet immobilization and washing, a streptavidin–phycoerythrin conjugate (SAPE) was used to illuminate the bound detection antibodies. Washed bead suspensions were then read using a FLEXMAP 3D^®^ analyzer (Luminex). The FLEXMAP 3D^®^ instrument contains two distinct diode lasers—a red laser (625 nm) that illuminates the unique internal fluorescent label to distinguish individual bead populations (classification channel) and a green laser (532 nm) that evaluates SAPE-associated fluorescence to quantify the captured biomarker analyte in each bead set. The median fluorescent intensity (MFI) of each bead set is directly proportional to the concentration of the bound analyte. Standard curves for each biomarker were generated on each assay plate using protein standards included in the Oncuria kit. All samples and standards were tested in duplicate.

*Statistics:* Descriptive statistics, including % detectable, % of control, minimum median fluorescent intensity (MFI), maximum MFI, and %CV, were reported as appropriate. Student’s t-test was used to compare continuous variables, and Fisher’s exact test was used to evaluate categorical variables. SAS V9.4 (SAS Institute Inc., Cary, NC, USA) and GraphPad Prism 10.0 (GraphPad Software Inc., La Jolla, CA, USA) software were used to perform statistical analyses. Two-tailed *p*-values < 0.05 were considered statistically significant.

## 3. Results

In the time-and-temperature study, we monitored Oncuria’s 10 analytes after storage at room temperature, 4 °C, and −20 °C over time in a pooled urine sample from five healthy individuals. Appendix A shows the mean ± SD concentrations of the analytes in the control subjects. Urine samples stored at room temperature resulted in a trend toward a time-dependent increase in the measured concentrations of MMP9, APOE, ANG, and MMP10 (Figure 1A). This effect of urine storage on the analytes was largely abrogated by storage at cooler temperatures; little-to-no change in the levels of any of the 10 assay analytes was observed in urine samples stored for up to 1 week at 4 °C (Figure 1B) and through 1 month at −80 °C (Figure 1C). Thus, storage at cooler temperatures largely stabilized the measured concentrations of all 10 Oncuria analytes. When the analyte concentrations were incorporated into the logistic regression algorithm to generate a bladder cancer risk score, the risk score for room temperature conditions ranged from 0.313 to 0.330, while the risk scores for 4 °C and −20 °C ranged from 0.247 to 0.313 and from 0.262 to 0.318, respectively. Thus, although the room temperature storage sample was associated with a modestly increased risk score, the risk score ranges at all three storage temperatures overlapped and are likely not statistically, or clinically, significant. These data still support the selection of 4 °C as an appropriate storage temperature for urine samples that will be evaluated with the Oncuria assay.

To explore the hypothesis that biospecimen stabilizers could prevent changes in urine sample analyte levels, thereby allowing storage at room temperature, we tested the performance of two commercially available stabilizers, cOmplete™ Protease Inhibitor Cocktail and Alere NMP22 Urine Collection Kit (Figure 2). The addition of either urine stabilizer to samples immediately after collection did not prevent the modest changes in Oncuria-determined levels of some analytes that occurred during storage at room temperature (Figure 2). In general, variation in measured analyte concentrations (e.g., A1AT) versus control levels appeared to be greater with the Alere NMP22 kit stabilizer than the cOmplete™ protease-inhibiting formulation.

The commercially available Urine Collection and Preservation Tube (Norgen Biotek) claims to prevent nucleic acid and protein degradation for up to 2 years in urine samples stored at room temperature according to the product insert. We evaluated the levels of the 10 Oncuria assay bladder cancer analytes in urine stored in these urine-stabilizing tubes and in standard sterile 15 mL conical tubes in common laboratory use (Figure 3). We were unable to discern any clear benefit of the urine-stabilizing container over a standard laboratory conical tube with respect to the consistency of Oncuria-measured biomarker levels after urine storage through 24 h at room temperature or 4 °C.

Next, we speculated that bacterial overgrowth at the time of voided urine collection could cause biospecimen instability at room temperature. Therefore, we tested the effect of trimethoprim, an antifolate antibiotic often used to treat urinary tract infections, on the stability of Oncuria analytes under various storage conditions (Figure 4). The addition of trimethoprim to urine samples did not appear to uniformly improve consistency in a subsequent analyte measurement in specimens stored at room temperature and at 4 °C for up to 24 h.

## 4. Discussion

Previously, we performed extensive validation studies [4,5] with Oncuria using refrigerated or frozen urine samples, demonstrating that the proposed procedure is suitable for its intended purpose. In the clinical setting, though refrigerating or freezing a urine sample is feasible, it may pose some challenges and impede testing adoption. Thus, we would like to streamline specimen collection and storage by avoiding the need for refrigerating or freezing the urine samples. Therefore, we set out to determine if technical variation due to sample collection and storage (i.e., not storing at 4 °C) could potentially influence the results of Oncuria as well as determine if stabilizers could mitigate these influences.

In this study, we first evaluated the effect of the storage temperature on the Oncuria’s test results. In study 1, we employed pooled urine samples from healthy volunteers. The results demonstrate that the storage of the urine samples at room temperature for >120 min significantly increased the levels of MMP9, APOE, ANG, and MMP10 (Figure 1A), which could change the final results (positive or negative to cancer). The diagnostic results from the sample stored in an uncontrolled temperature can be misleading when they are incorrect or false, which can lead to misguided treatments and harm patients. On the other hand, urine samples stored at 4 °C and −20 °C demonstrated no significant changes, suggesting that refrigeration and freezing stabilize urine samples (Figure 1B,C). As noted above, when the analyte concentrations were incorporated into the logistic regression algorithm associated with this test, there were no clinically significant changes in the algorithm output. Though encouraging, significant concern arises with the changes in analyte concentrations at room temperature. Therefore, while exercising extreme caution, we are advocating for storing Oncuria urine samples at 4 °C.

In studies 2 and 3, we tested the performance of protease inhibitors on the stability of urine samples. The results demonstrate that all tested stabilizers were not able to stabilize the levels of the 10 protein analytes. Noteworthy, the results confirm that the current storage of voided urine samples at 4 °C is the best strategy to maintain the integrity of the proteins (Figure 1, Figure 2 and Figure 3). This aligns with a previous study evaluating the stability of urine specimens for urinalysis, which noted that the voided urine samples must be refrigerated after collection until analyzed to maintain integrity.

It is speculated that the refrigeration slows the growth of bacteria introduced into the urine from the urethra and/or prepuce/labia at the time of urine collection. The overgrowth of bacteria may directly or indirectly affect the 10 protein analytes. Therefore, in study 4, we utilized an antibiotic as a biospecimen stabilizer. Once again, the additive was not effective in stabilizing the results of Oncuria at room temperature (Figure 4). Taking all results together, we confirmed that the storage of voided urine samples at 4 °C is the best strategy to maintain the urine sample quality.

The current study has several limitations. First, due to the volume of urine needed to analyze the various conditions, we employed healthy controls, which were able to provide at least 300 mL. Second, due to the various conditions, we employed only 1–2 subjects in studies 2 through 4. Third, we only utilized commercially available proteinase inhibitors and urine stabilizers, i.e., we did not test new potential stabilizers. Finally, as participants may have speculated the nature of the voided urine samples, their collection methods may have been more astute than is commonly seen in the average clinical setting.

Urine collection in the clinic is noted to have intrinsic variability. This variability can be minimized via refrigeration, not through the addition of biospecimen stabilizers or antibiotics to voided urine samples immediately after collection. Altogether, we recommend placing the mid-stream voided urine samples for Oncuria in refrigeration alone within 2 h of collection for subsequent testing.

## 5. Conclusions

To our knowledge, this is the first multiplex bead-based immunoassay that has undergone extensive analytical testing to ensure appropriate analytical and clinical performance. Herein, we noted that voided urine samples must be refrigerated to maintain the integrity of the sample for clinical testing with Oncuria, which is important to prevent misguided treatments and harm patients.

## Figures and Tables

**Figure 1 diagnostics-15-00138-f001:**
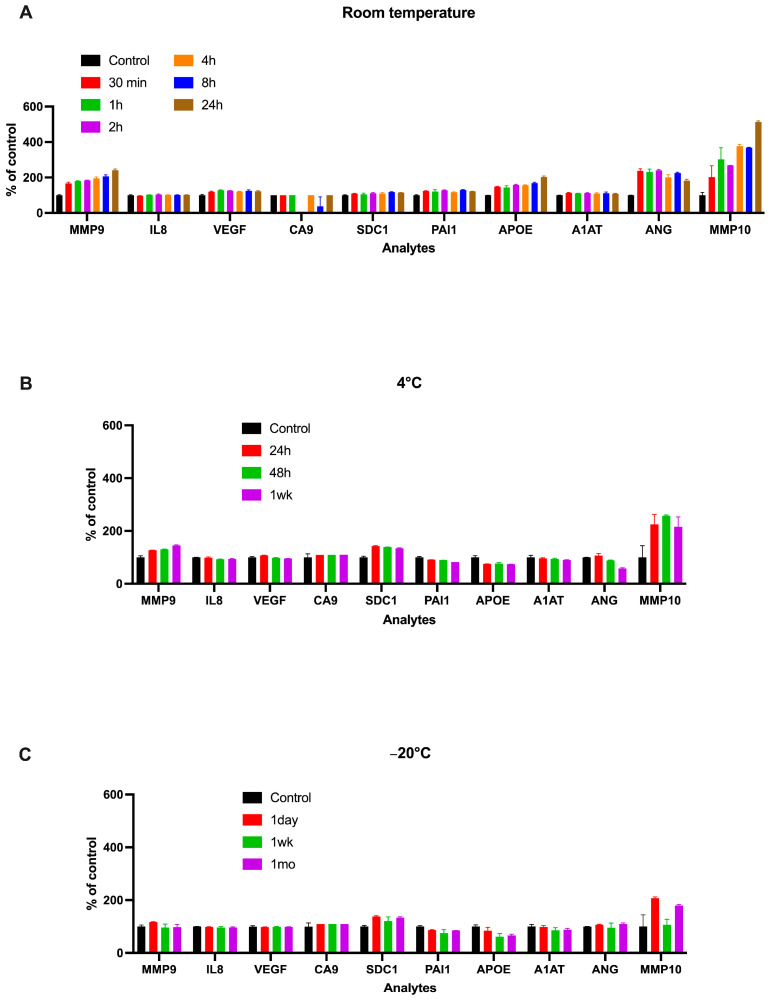
**Effect of storage temperature and time on 10 bladder cancer biomarker levels in human urine specimens.** Concordant evaluation of 10 urinary analytes from the Oncuria bladder cancer assay in pooled urine stored at room temperature (**A**), 4 °C (**B**), and −20 °C (**C**). Urine samples kept at room temperature increased the assayed levels of MMP9, APOE, ANG, and MMP10 over time. The results suggest that keeping the sample in an uncontrolled temperature causes unreliable test results. Error bars indicate the standard deviation of the mean.

**Figure 2 diagnostics-15-00138-f002:**
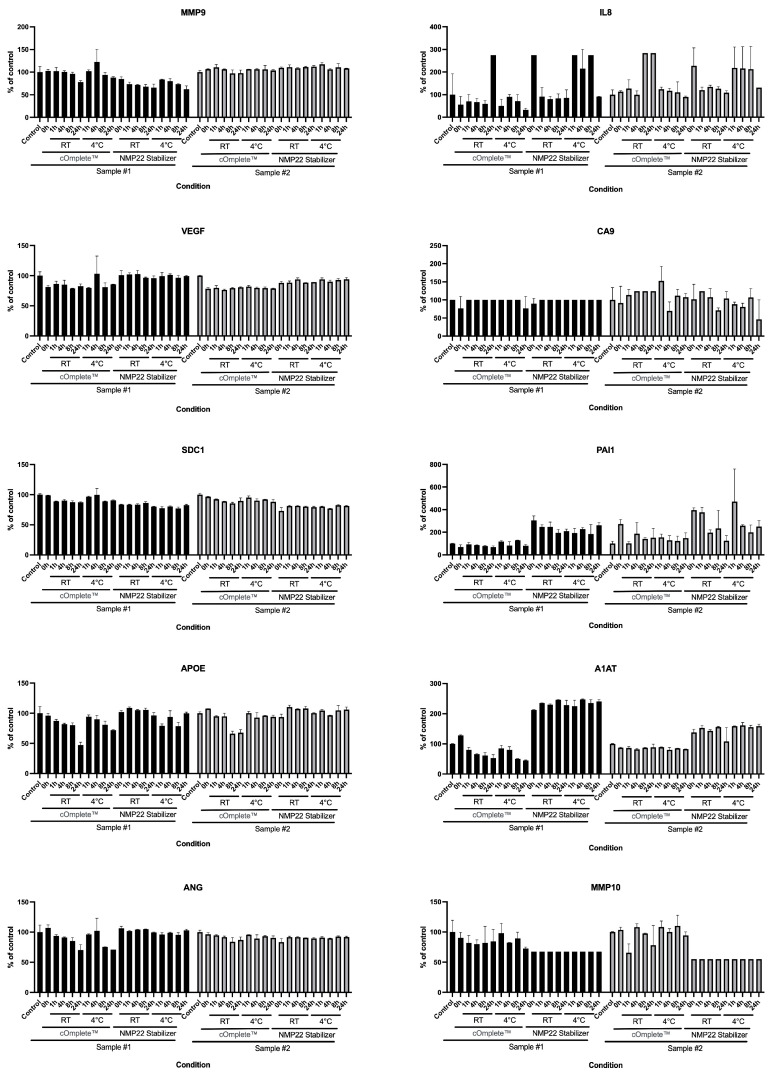
**Effect of commercial preservative formulations on 10 bladder cancer biomarker levels in human urine specimens**. The addition of neither cOmplete™ Protease Inhibitor Cocktail nor the urine stabilizer provided in the Alere NMP22 Urine Collection Kit prevented changes in Oncuria-measured levels of the 10 urinary analytes in urine, such as decreased levels of analytes including MMP9, SDC1,APOE, A1AT, ANG, and MMP10 when samples from two individuals were stored at room temperature or at 4 °C for up to 24 h. In addition, the NMP22 stabilizer increased the levels of IL8, PAI1, and A1AT. The results suggest that these preservatives did not stabilize some analytes but reduced the accuracy of this assay. Error bars indicate the standard deviation of the mean.

**Figure 3 diagnostics-15-00138-f003:**
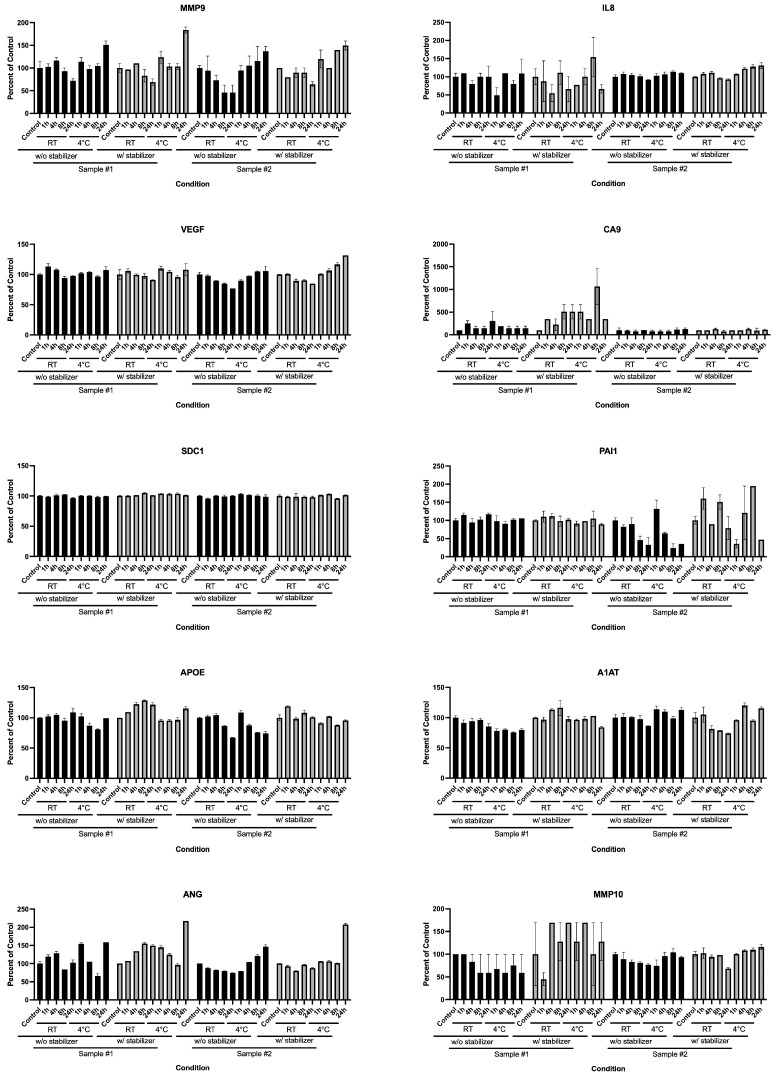
**Effects of a commercial Urine Collection and Preservation Tube on 10 bladder cancer biomarker levels in human urine specimens.** Analytes were evaluated in urine samples using the Oncuria multiplex bladder cancer assay after sample storage in commercial Urine Collection and Preservation Tubes or in typical sterile 15 mL conical tubes, at room temperature and at 4 °C, for up to 24 h. There was no clear difference in all analytes between urines in preservation tubes and typical sterile 15 mL conical tubes. Error bars indicate the standard deviation of the mean.

**Figure 4 diagnostics-15-00138-f004:**
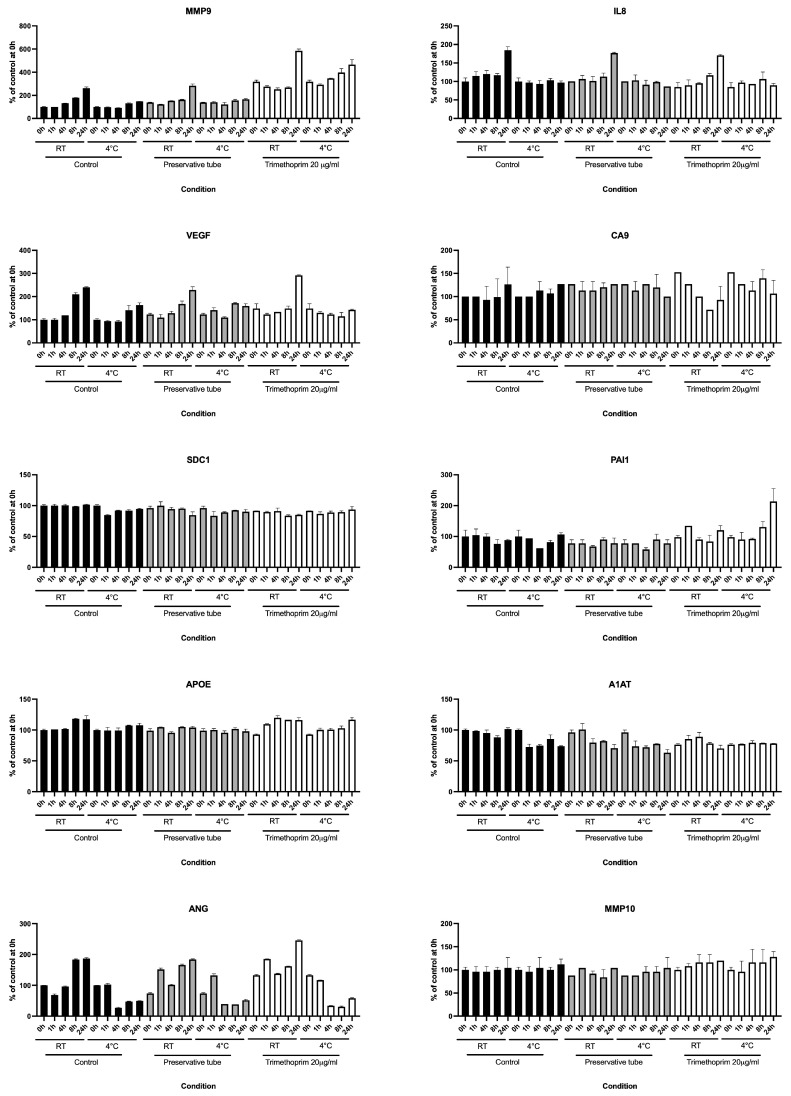
**The effects of Urine Collection and Preservation Tube and trimethoprim (final concentration of 20 μL/mL) on 10 urinary analytes from Oncuria in pooled urine kept at room temperature and at 4 °C for up to 24 h.** There was no clear difference in all analytes between urines in typical sterile 15 mL conical tubes and preservation tubes and urine with antibiotics (trimethoprim). Error bars indicate the standard deviation of the mean.

## Data Availability

All data are found in this manuscript.

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
