# Peer review of "Effects of Different Voided Urine Sample Storage Time, Temperature, and Preservatives on Analysis with Multiplex Bead-Based Oncuria Bladder Cancer Immunoassay"

_diagnostics, 2025, doi:10.3390/diagnostics15020138_

Round 1

Reviewer 1 Report

Comments and Suggestions for Authors

It is an interesting internal control study to analyte those markers level in collection of voiding urine. The author want to identify the well status for analysis with multiple bead-based Oncuria bladder cancer immunoassay.   But in this article, some questions were needed to clarity.

1.      In this article, the authors make the conclusion to prevent storage of urine at room temperature. The all figures present those individual results in room temperature and 4oC. Therefore, the results from -20oC was not seen. Please add them.

2.      Actually, the voiding urine was full with many materials, as inflammatory cell or infection cell. The author just analyte urine from health people, and it was not enough to provide the results to the cancer patients. If had experimental data, please add the results from UTI patents or even cancer patients.

Reviewer 2 Report

Comments and Suggestions for Authors

The manuscirpt titled “Effect of Different Voided Urine Sample Storage Time and Temperature, and Preservatives, on Analysis with the Multiplex Bead-Based Oncuria Bladder Cancer Immunoassay” by Furuya,  this study assessed the impact of storage conditions on the stability of urine samples analyzed with the Oncuria bladder cancer immunoassay, which quantifies 10 protein biomarkers. Results showed that storage at room temperature led to significant alterations in biomarker levels, even with commercial stabilizers. Refrigeration at 4℃ was found to be essential for maintaining sample integrity, ensuring accurate analysis and minimizing degradation. This is a novel aspect in a field of intense research. Still some issues need to be clarified, as listed below, before the manuscript can be accepted for publication in Diagnostics.

1. The use of only two healthy individuals in Studies 2 and 3 raises concerns regarding the generalizability of the findings. The limited sample size may not adequately capture variability in biomarker stability across a broader population. It is recommended that the authors consider including a more diverse cohort, incorporating variations in age, gender, and underlying health conditions, to validate the effects of storage conditions on urine sample stability and enhance the applicability of their conclusions.

2. The manuscript notes significant increases in certain analytes, such as MMP9 and MMP10, at room temperature. It would enhance the study's impact if the authors provided a more detailed discussion of the potential biological or chemical mechanisms underlying these observed changes. This could include relevant literature or hypotheses related to the stability or activity of these analytes under the experimental conditions.

3. The methods used to evaluate the statistical significance and the variability represented by the error bars in the results should be clarified. Additionally, information on the number of replicates performed for each experiment would strengthen the readers' confidence in the data reliability. If replication was limited, the authors should address how this might impact the robustness of their findings.

4. While the manuscript recommends refrigeration as the optimal storage method, it would be valuable for the authors to discuss practical alternatives suitable for resource-limited settings, such as those encountered in developing regions or field studies. Exploring these alternatives could broaden the applicability of the study's findings.

5. The discussion on how analyte stability impacts diagnostic accuracy could be strengthened by quantifying the clinical error margins observed between refrigerated and room temperature storage. Providing specific metrics would help contextualize the clinical significance of the differences and enhance the translational relevance of the study.

6. The presentation of Figure 1 could be improved by ensuring that panels a, b, and c have consistent widths and are properly aligned. This adjustment would enhance the visual clarity and professionalism of the figure.

Round 2

Reviewer 1 Report

Comments and Suggestions for Authors

The revised version has enough information to support the original design study.

Reviewer 2 Report

Comments and Suggestions for Authors

Thank you for revising the manuscript accordingly. As the assigned reviewer, I have reviewed the revised manuscript and am pleased to inform you that I find the modification to be satisfactory. Therefore, I am pleased to recommend the acceptance of the revised manuscript for publication.